# Continual learning with the neural tangent ensemble

**Ari S. Benjamin**      **Christian Pehle**      **Kyle Daruwalla**
Cold Spring Harbor Laboratory
Cold Spring Harbor, NY 11724
{benjami,pehle,daruwal}@cshl.edu

## Abstract

A natural strategy for continual learning is to weigh a Bayesian ensemble of fixed functions. This suggests that if a (single) neural network could be interpreted as an ensemble, one could design effective algorithms that learn without forgetting. To realize this possibility, we observe that a neural network classifier with N parameters can be interpreted as a weighted ensemble of N classifiers, and that in the lazy regime limit these classifiers are fixed throughout learning. We call these classifiers the *neural tangent experts* and show they output valid probability distributions over the labels. We then derive the likelihood and posterior probability of each expert given past data. Surprisingly, the posterior updates for these experts are equivalent to a scaled and projected form of stochastic gradient descent (SGD) over the network weights. Away from the lazy regime, networks can be seen as ensembles of adaptive experts which improve over time. These results offer a new interpretation of neural networks as Bayesian ensembles of experts, providing a principled framework for understanding and mitigating catastrophic forgetting in continual learning settings.

## 1   Introduction

Neural networks often forget previous knowledge when trained with gradient descent. In contrast, animals learn from sequential experiences, suggesting that true 'lifelong learners' use different strategies for learning [25].

One strategy to learns without forgetting is to update the posterior distribution over a set of fixed probabilistic models [8]. This includes any fully Bayesian model, such as Bayesian linear regression. The fundamental reason why these algorithms do not forget information is because the posterior over models is invariant to data sequence. Given two permutations of the data, the posterior will be the same. This property of posteriors has inspired many strategies to reduce forgetting by approximating the posterior distribution over neural networks [22, 24, 11, 28, 26, 41, 37]. However, these approximations introduce many new parameters and considerable memory overhead. In general, estimating the full posterior distribution over high-dimensional networks is prohibitive.

Here, we shift our perspective and instead interpret a *single* neural network as an ensemble of many experts. This allows tracking a posterior (over experts, instead of networks) without introducing any memory overhead besides the network itself.

This motivates our main result, which we note is generally applicable outside of continual learning. More specifically, we show that **neural network classifiers perturbed by a small vector in parameter space can be described as a weighted ensemble of valid classifiers** outputting a probability distribution over labels. We call this the Neural Tangent Ensemble (NTE). Inspired by the Neural Tangent Kernel, this result depends on a first-order Taylor expansion around a seed point [19]. As a consequence, it operates as an ensemble of *fixed* classifiers in the NTK limit of infinite width.

In this framework, learning is framed as Bayesian posterior updating rather than optimization. These two approaches might be expected to be quite different, as a posterior update is multiplicative whereas gradient-based optimization is additive. Surprisingly, however, we find that the NTE posterior update rule is approximately stochastic gradient descent (SGD) on the network with batch size 1, thus shedding new light on the dynamics of neural network optimization.

Our primary contributions are:

- We introduce the Neural Tangent Ensemble (NTE), a novel formulation that interprets networks as ensembles of classifiers, with each parameter contributing one classifier.
- We derive the posterior update rule for the NTE for networks in the lazy regime, in which experts are fixed, and show that it is equivalent to single-example stochastic gradient descent (SGD) without momentum, projected to the probability simplex.
- This justifies the empirical finding that SGD with no momentum forgets much less than standard optimizer settings.
- We demonstrate that catastrophic forgetting in neural networks is associated with the transition from the lazy to the rich regime.

## 2 Motivation: Ensembles are natural continual learners

To demonstrate why Bayesian ensembles make good continual learners, consider a function $f(x)$ that is an ensemble of many experts $f_i(x)$ (Fig. 1). We will consider what it takes to modify this ensemble so that it performs well on two tasks $\mathcal{A}$ and $\mathcal{B}$.

A simple strategy for continual learning is to prune away experts. Let $\mathcal{S}_\mathcal{A}$ be the set of functions that are good (and equally good) for task $\mathcal{A}$. A good ensemble can be constructed by sampling from $\mathcal{S}_\mathcal{A}$:

$$f_\mathcal{A}(x) = \frac{1}{N} \sum_{f_i \in \mathcal{S}_\mathcal{A}} f_i(x).$$

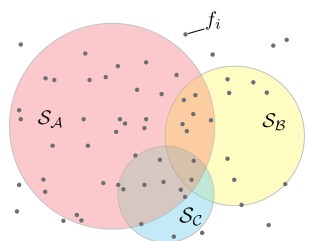

Figure 1: High-level intuition for model averaging and continual learning. Pruning the set of functions $f_i$ to those good for task $\mathcal{A}$, followed by further pruning for tasks $\mathcal{B}$ and $\mathcal{C}$, will result in a set of $f_i$ still good on $\mathcal{A}$.

Given a subsequent task $\mathcal{B}$, a new ensemble can be constructed on the fly by continuing to prune away the experts in $f_\mathcal{A}(x)$ that perform poorly on task $\mathcal{B}$. The remaining ensemble is still composed of experts from $\mathcal{S}_\mathcal{A}$ (assuming that the set intersection is not zero).

In contrast to many continual learning strategies for neural networks, this does not require replay, task boundaries, or any additional memory dedicated to preserving old task performance.

### 2.1 Belief updating generalizes set intersections

In real ensembles, experts do not perform equally well. This justifies weighing each expert in the ensemble with weights $p_i$ which are chosen such that $\sum_i^N p_i = 1$:

$$f_\mathcal{A}(x) = \sum_{f_i \in \mathcal{F}} p_i f_i(x). \tag{1}$$

This is particularly convenient if the experts encode the probability or belief about an event, $f_i(x) = p(y|x, f_i)$. In this case, one can weigh each function by its posterior probability given previous data:

$$p(y|x, \mathcal{D}) = \sum_{f_i \in \mathcal{F}} p(f_i|\mathcal{D}) \, p(y|x, f_i). \tag{2}$$

This is the optimal weighing strategy when the experts can be assumed to be independent [45].

It is useful to contrast the ensemble in Eq. 2 with linear regression using a feature map, $f(x) = \sum_i w_i \phi_i(x)$, as might be observed in kernel regression. In an ensemble the weights $w_i$ are strictly positive, whereas weights in regression may switch sign arbitrarily.

## 2.2 The posterior is invariant to data ordering

The property of Bayesian ensembles that motivates this paper is that the posterior probability of each expert is invariant to the order in which data in seen. This is because, like set intersections, single-task posteriors multiply to form the multi-task posterior:

$$p(f_i | \mathcal{A} \cap \mathcal{B}) \propto p(f_i | \mathcal{A}) p(f_i | \mathcal{B}). \tag{3}$$

This property is restated more formally in the following Lemma:

**Lemma 1.** *Invariance to data ordering in Bayesian Ensembles. Let $\mathcal{F} = f_1, ..., f_N$ be a set of fixed experts, $\mathcal{W} = w_1, ..., w_N$ be their weights, and $\mathcal{D} = D_1, ..., D_T$ be a sequence of datasets from $T$ tasks. Then, for any permutation $\pi$ of the indices $1, ..., T$, $p(f_i | \mathcal{D}) = p(f_i | D_1, ..., D_T) = p(f_i | D_{\pi(1)}, ..., D_{\pi(T)})$*

*Proof.* By Bayes' rule, $p(f_i | \mathcal{D}) \propto p(f_i) \prod_{t=1}^{T} p(D_t | f_i)$. The right-hand side is a product of terms, one for each dataset. Since multiplication is commutative, $\prod_{t=1}^{T} p(D_t | f_i) = \prod_{t=1}^{T} p(D_{\pi(t)} | f_i)$ for any permutation $\pi$. Therefore, $p(f_i | D_1, ..., D_T) = p(f_i | D_{\pi(1)}, ..., D_{\pi(T)})$. $\qquad\square$

Thus, there is no catastrophic forgetting problem for models which are ensembles of fixed, independent probabilistic classifiers. This motivates assessing under what conditions neural networks approach this setting.

## 3 The Neural Tangent Ensemble

How might a neural network be described as an ensemble? One simple strategy would be to take the last network layer as a set of functions, and then to choose the output weights according to their relative performance. However, this is is an expensive strategy to construct a relatively small set of classifiers, and it does not specify how earlier weights might change.

Here, we employ a first-order Taylor expansion to show that neural networks are (approximately) large ensembles over $N$ component functions, one for each edge in the network. We will examine a neural network $p(y|x, W^{(t)})$ with parameters $W^{(t)}$ whose output represents the probability or confidence of a label $y$ given input $x$. We can describe this output with a linearization around a very nearby *seed point* $W^{(0)}$. Note that we use the notation $W^{(0)}$ and $W^{(t)}$ for consistency and in general $W^{(0)}$ need not be the initialization or on the optimization trajectory at all.

$$p(y|x, W^{(t)}) \approx p(y|x, W^{(0)}) + \sum_{i}^{N} \Delta w_i \frac{\partial p(y|x, W^{(0)})}{\partial w_i^{(0)}} \tag{4}$$

At first glance it does not appear that this Taylor expansion is an ensemble. There seem to be no true classifiers: the gradients are not probabilities over classes $y$, being neither nonnegative, bounded, nor normalized to 1 across the output labels. Nor are there true weights, as $\Delta w_i$ is also not nonnegative. However, both of these criteria can be met with some rearrangements and with the assumption that the loss is sufficiently smooth with respect to its parameters. This leads to our main result:

**Theorem 2.** *Suppose $p(y|x, W^{(0)})$ is a neural network for which the log-likelihood is $L-$Lipschitz continuous in its parameters, i.e. all gradients of the loss are bounded by a constant $L$. Let $W^{(0)}$ then be perturbed by a $\Delta W$ with $\|\Delta W\|_1 = z$. If the perturbation is sufficiently small (with $zL < 1$), then **the network can be described as an ensemble of a set of N classifiers** $\{p(y|x, f_i)\}_i^N$, each with weight $\frac{|\Delta w_i|}{z}$, plus higher-order contributions which vanish for small $z$:*

$$p(y|x, W^{(t)}) = \sum_{weights\ i}^{N} \frac{|\Delta w_i|}{z} \, p(y|x, f_i) \, + \, \mathcal{O}(\|\Delta W\|_2^2)$$

*Each classifier $p(y|x, f_i)$, which we call the **neural tangent expert**, outputs a probability distribution over labels y:*

$$p(y|x, f_i) = p(y|x, W^{(0)}) \left( 1 + z \, sign(\Delta w_i) \frac{\partial}{\partial w_i^{(0)}} \log p(y|x, W^{(0)}) \right)$$

The proof is postponed to Appendix 8.1. Informally, it relies two simple rearrangements: splitting the weights into sign and magnitude $\Delta w_i = |\Delta w_i| sign(\Delta w_i)$, and bringing the zeroth order term inside the first-order sum. This results in a sum over a term which, surprisingly, sums to 1 over the output labels and is weighted by a term that sums to 1 over experts, meeting the criteria of an ensemble.

This simple reformulation invites a change in perspective about the role of each parameter in a deep neural network. Each parameter contributes a separate classifier. The distributed architecture and connected paths of the network matter, but they explicitly contribute through the gradients alone.

In the literature on ensembles, a common focus is to examine the *quality* and *diversity* of the experts separately. By the bias/variance decomposition, both aspects enter in the generalization error [38, 47]. Here, it is clear that all experts share a factor that is the overall quality of the center of the Taylor expansion, $p(y|x, W^{(0)})$. What distinguishes experts from one another is the diversity of network gradients.

## 3.1 Experts are fixed in the lazy regime

This paper is motivated by the fact that Bayesian ensembles of *fixed* experts do not forget past data when learning by posterior updating. Under what conditions is the Neural Tangent Ensemble composed of fixed functions?

The answer to this question follows directly from the literature on the Neural Tangent Kernel (NTK) and lazy regime networks [19, 7]. If the network is in the 'lazy' regime, then the Jacobian of the network does not change during gradient descent learning and the linearization remains valid. This occurs in the limit of infinite width for MLPs for certain initializations [19]. (Output scaling also controls laziness [7], and is a necessary when using softmax nonlinearities even in the infinite width [29].) As a consequence, the experts in the NTE interpretation are fixed functions in the lazy regime.

## 3.2 Learning ensemble weights

If a network is secretly an ensemble, how should it learn from new data? The natural next step is to convert the NTE into a Bayesian ensemble. In a Bayesian ensemble, the weight of each function is its posterior probability given past data:

$$\frac{|\Delta w_i|}{z} \leftarrow p(f_i|\mathcal{D}) = \frac{p(\mathcal{D}|f_i) \, p(f_i)}{\sum_i p(\mathcal{D}|f_i) \, p(f_i)}. \tag{5}$$

This can be seen as the E step in a generalized EM algorithm [45]. In the following section, we will describe how to calculate this posterior probability with an online learning algorithm. For the moment, we assume the experts are fixed functions (i.e. the network is lazy).

### 3.2.1 The data likelihood

**Lemma 3.** *For IID data $\mathcal{D} = \{x_k, y_k\}_{k=1}^P$, the likelihood of the data under an expert can be written in terms of a log-likelihood loss function $\ell_k^{(0)} = -\log p(y_k|x_k, W^{(0)})$ of the network at initialization:*

$$p(\mathcal{D}|f_i) = \prod_{examples \, k} e^{-\ell_k^{(0)}} \left( 1 - z \, sign(\Delta w_i) \frac{\partial}{\partial w_i^{(0)}} \ell_k^{(0)} \right) \tag{6}$$

*Proof.* Starting with the data likelihood,

$$p(\mathcal{D}|f_i) = \prod_{\text{examples } k} p(y_k|x_k, f_i) \tag{7}$$

$$= \prod_{\text{examples } k} \left( p(y_k|x_k, W^{(0)}) + z \operatorname{sign}(\Delta w_i) \frac{\partial}{\partial w_i^{(0)}} p(y_k|x_k, W^{(0)}) \right) \tag{8}$$

$$= \prod_{\text{examples } k} p(y_k|x_k, W^{(0)}) \left( 1 + z \operatorname{sign}(\Delta w_i) \frac{\partial}{\partial w_i^{(0)}} \log p(y_k|x_k, W^{(0)}) \right) \tag{9}$$

Plugging in the definition of $\ell_k^{(0)}$ yields the above expression. $\qquad\square$

### 3.2.2 The posterior probability: renormalization

After the data likelihoods are computed for each neural tangent expert, they must be renormalized to obtain the posterior probabilities. In our case, we naturally have access to a very large number of tangent experts and their likelihoods. Indeed, if the width is indeed taken to infinity, this there are infinitely many neural tangent experts in a single network. We propose to use the distribution of likelihoods in the current network as a Monte Carlo estimate of the normalizing constant.

$$p(f_i|\mathcal{D}) = \frac{\prod_{\text{examples } k} e^{-\ell_k^{(0)}} \left( 1 - z \operatorname{sign}(\Delta w_i) \frac{\partial}{\partial w_i^{(0)}} \ell_k^{(0)} \right) p(f_i)}{\sum_i \prod_{\text{examples } k} e^{-\ell_k^{(0)}} \left( 1 - z \operatorname{sign}(\Delta w_i) \frac{\partial}{\partial w_i^{(0)}} \ell_k^{(0)} \right) p(f_i)} \tag{10}$$

This can be simplified by noting that each $e^{-\ell_k^{(0)}}$ term will cancel; the product $\prod_k e^{-\ell_k^{(0)}}$ appears in every additive term in the denominator. Assuming a uniform prior $p(f_i)$, we then have:

$$p(f_i|\mathcal{D}) = \frac{\prod_k \left( 1 - z \operatorname{sign}(\Delta w_i) \frac{\partial}{\partial w_i^{(0)}} \ell_k^{(0)} \right)}{\sum_i \prod_k \left( 1 - z \operatorname{sign}(\Delta w_i) \frac{\partial}{\partial w_i^{(0)}} \ell_k^{(0)} \right)} \tag{11}$$

### 3.3 The posterior update is (almost) stochastic gradient descent

We will now link this posterior expression with a neural network update rule. Recall that in Theorem 2, the normalized magnitude of each perturbation is interpreted as the posterior probability of the corresponding neural tangent expert.

$$\frac{|\Delta w_i|}{z} \leftarrow p(f_i|\mathcal{D})$$

This means the parameters can act as a running cache of the posterior as new data is encountered. As in standard belief updating, this involves a likelihood update followed by renormalization. Surprisingly, this multiplicative belief update rule yields an update which is very close to SGD.

**Lemma 4.** *For any network that is well-described as a first-order Taylor expansion around around $W^{(0)}$ with perturbation $\|\Delta W\|_1 = z$, the posterior belief update given a new example is equivalent to single-example stochastic gradient descent under a cross-entropy loss objective, subject to the constraint that $\|\Delta W\|_1 = z$, and using a per-parameter learning rate of $z|\Delta w_i|$.*

*Proof.* The proof is a matter of writing out how the posterior changes with a single example. Multiplying by the likelihood of a new example, the unnormalized posterior updates as:

$$\frac{|\Delta w_i'^{(t)}|}{z} = \frac{|\Delta w_i^{(t-1)}|}{z} \left( 1 - z \operatorname{sign}(\Delta w_i^{(t-1)}) \frac{\partial}{\partial w_i^{(0)}} \ell_k^{(0)} \right) \tag{12}$$

This multiplicative update for the unnormalized weights can also be written an *additive* rule. Multiplying by $z$ and by $\text{sign}(\Delta w_i)$,

$$\Delta w_i'^{(t)} = \Delta w_i^{(t-1)} - z|\Delta w_i^{(t-1)}|\frac{\partial}{\partial w_i^{(0)}}\ell_k^{(0)} \tag{13}$$

One can add the initial parameters to either side to yield a rule in the space of network parameters:

$$w_i'^{(t)} = w_i^{(t-1)} - z|\Delta w_i^{(t-1)}|\frac{\partial}{\partial w_i^{(0)}}\ell_k^{(0)} \tag{14}$$

This is true (single-example) stochastic gradient descent *projected in the L1 diamond* with a learning rate $z|\Delta w_i|$. Note that this does not allow averaging gradients across examples (a "batch size of 1" update) and that it uses the gradients at initialization (though see section 4.1).

To complete the update, the parameters should then be renormalized such that $\sum_i |\Delta w_i^{(t)}| = z$.

An alternative normalization scheme is to use a gradient projection algorithm. Adding a Langrage multiplier $\gamma$ to Eq. 13 and solving for the $\gamma$ that ensures $\sum_i |\Delta w_i| = z$ yields a update which keeps $\|\Delta W\|_1 = z$ even without renormalization:

$$w_i^{(t)} = w_i^{(t-1)} - z\left(|\Delta w_i^{(t-1)}|\frac{\partial}{\partial w_i^{(0)}}\ell_k^{(0)} - \text{avg}_j\left(|\Delta w_j^{(t-1)}|\frac{\partial}{\partial w_j^{(0)}}\ell_k^{(0)}\right)\right) \tag{15}$$

$\square$

Not only is the posterior update tractable, then, but it is sufficiently close to gradient descent that it can be interpreted in a standard optimization framework.

Although it may seem that our result would depend on the idiosyncratic likelihood function of the NTE, this result is nevertheless similar to previous algorithms that have been proposed as ways to weigh many experts. At high level, our result appears similar to the Multiplicative Weights algorithm described in [1]. Another interpretation of this algorithm is as the *approximated exponential gradient descent with positive and negative weights* algorithm from [23] but applied to the change in weights $\Delta W$. There, it is derived by minimizing an arbitrary loss function under a constrained change in the relative entropy over ensemble weights to obtain the *exponentiated gradient descent algorithm*, which is then linearized with a Taylor expansion in the approximated version.

### 3.4   Summary of the NTE theory

The Neural Tangent Ensemble is an interpretation of networks as ensembles of *neural tangent experts*. Updating the NTE of lazy networks as a Bayesian ensemble creates a perfect continual learner, in the sense that the multitask solution is guaranteed to be the same as the sequential task solution.

The posterior probability of each expert in the NTE is surprisingly tractable. Given a new example, the update rule is a simple additive rule in the space of network parameters which can be interpreted as projected gradient descent scaled by the change in parameters since initialization.

## 4   Networks away from the lazy regime

In real finite-width networks, gradients change throughout learning. Since each weight's corresponding neural tangent expert changes, there is no guarantee that weights at time $t$ still reflect the cumulative likelihood of past data under that expert.

This phenomenon is clearly observed empirically by measuring how much experts change under the NTE update rule as a function of network width. In Fig. 2, we measure the average change in expert's Jacobian from initialization after training on MNIST as a function of network width with the NTE rule described above. Experts change less in wider networks than in narrow networks.

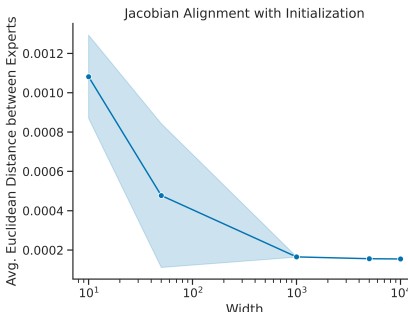

Figure 2: The average squared difference between experts' columns of the Jacobian measured at initialization and the end of training on MNIST with an 2-layer ReLU MLP and the NTE rule. Error bands indicate the standard deviation over 10 random seeds. As the width of the network increases, the average distance decreases, indicating the larger networks remain closer to the original linearization.

Another way this can be measured is by verifying that, in finite-width networks, the posterior update rule using the gradients around initialization does not lead to effective training. In Figure 3, we confirm that as the gradients lose correlation with the gradient at initialization, performance begins to rapidly degrade. This echoes the findings of [7] that linearized CNNs do not learn as effectively as their non-lazy counterparts. Thus, the NTE posterior update rule as written above is only effective when the Jacobian is truly static.

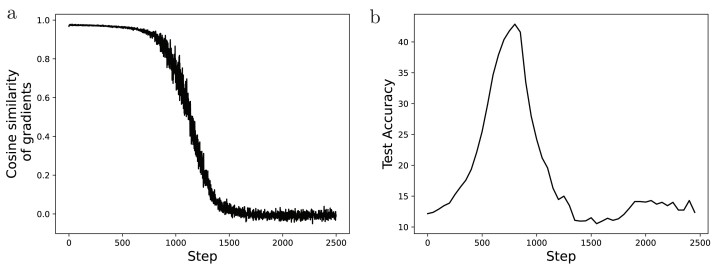

Figure 3: a) Gradients of an MLP at time $(t)$ rapidly lose correlation with the gradients at initialization. b) Training a network with the NTE posterior update rule fails when gradients diverge. Hyperparameters are reported in the Appendix.

## 4.1 Rich-regime networks are ensembles of adaptive experts

To ensure the NTE formulation remains valid, one can allow the seed point of the Taylor expansion (the 'initialization') to change throughout learning. This has an interesting interpretation. Namely, it allows us to view finite-width neural networks as **ensembles of changing neural tangent experts**.

**Lemma 5.** *(informal) Let $W^{(t)}$ be the parameters of a (finite-width) neural network. Choose a nearby **seed point** $\tilde{W}^{(t)}$ as $W^{(t)} + \epsilon$, with $\epsilon$ fixed and $\|\epsilon\|_2$ sufficiently small relative to the curvature such that the Jacobians of the log output probabilities of the perturbed and unperturbed networks are identical, $J(\tilde{W}^{(t)}) = J(W^{(t)})$. The network can then be written as an ensemble of adaptive experts:*

$$p(y|x, f_i^{(t)}) = p(y|x, \tilde{W}^{(t)}) \left(1 + \|\epsilon\|_1 sign(\epsilon_i) \frac{\partial}{\partial w_i^{(t)}} \log p(y|x, W^{(t)})\right)$$

*If $\epsilon$ is set as the uniform vector with values $\epsilon_i = \sqrt{\eta/N}$, the learning rate in the posterior update rule reduces to $\|\epsilon\|_1 |\epsilon_i| = \eta$ and we recover stochastic gradient descent with mean-centered gradients and learning rate $\eta$:*

$$w_i^{(t+1)} = w_i^{(t)} - \eta \left(\frac{\partial}{\partial w_i^{(t)}} \ell_K^{(t)} - avg_j \left(\frac{\partial}{\partial w_j^{(t)}} \ell_K^{(t)}\right)\right) \qquad (16)$$

Rich-regime learning is thus akin to a particle filter; each expert changes individually, but the prediction is the ensemble vote.

A interesting feature of this lemma is the equivalence between the rule that improves each expert (gradient descent on $w$) and the rule that decides how to weigh the experts in the ensemble (also gradient descent on $w$). This need not have been the case. As a result, one can perform belief updating assuming a fixed ensemble and end up improving each expert within it.

## 4.2 The NTE rule with current gradients

Motivated by this result, we evaluated how well the NTE posterior update rule works when the gradients evaluated at initialization, $\frac{\partial}{\partial w_i^{(0)}}\ell_K^{(0)}$, are replaced with the gradients of the current network $\frac{\partial}{\partial w_i^{(t)}}\ell_K^{(t)}$. These converge in the infinite-width limit.

To obtain a practical algorithm, we additionally modify the NTE update rule with two hyperparameters that control the learning rate. First, noting that $z$ in Eq. 14 acts as a learning rate, we replace it with a tunable parameter $\eta$. Secondly, we introduce a regularization parameter $\beta$ which keeps the network close to initialization as measured by the relative entropy of the change in parameters (see Appendix 8.2 for derivation). This constrains the amount of information contained in the weights [17].

Pseudocode for the resulting algorithm is in the Appendix 1. We also display the result of sweeps over $\beta$ and $\eta$ on the Permuted MNIST task in Fig. 7.

## 5 Predictions and results

Our findings suggest several ways to reduce forgetting in finite networks. First, networks closer to the lazy regime will better remember old tasks as long as the update rule is sufficiently similar to the NTE posterior update rule. Second, one should be able to reduce forgetting by ablating standard optimization methods like momentum and moving towards the NTE posterior update rule.

Below, we verify these predictions on the Permuted MNIST task with MLPs and on the task-incremental CIFAR100 with modern CNN architectures. In the Permuted MNIST task, an MLP with 10 output units is tasked with repeatedly classifying MNIST, but in each task the pixels are shuffled with a new static permutation. In task-incremental CIFAR100, a convolutional net with 100 output units sees only 10 classes each task. In the terminology of van de Ven et al, this is a task-incremental task, whereas Permuted MNIST is a domain-incremental task [48].

### 5.1 Momentum causes forgetting

Momentum is not appropriate in a posterior update framework because it over-counts the likelihood of past data. Furthermore, it is a history-dependent factor. By contrast, posterior update rules are multiplicative and give identical results regardless of the order of data presentation.

Here, we report that *any* amount of momentum with SGD is harmful for remembering past tasks. To our knowledge, this has not been noted by previous empirical studies on catastrophic forgetting [13, 36, 35, 2]. As can been seen in Fig. 4, increasing momentum monotonically increases forgetting a first task on Permuted MNIST. Similar trends exist for ResNet18 and ConvNeXtTiny on the CIFAR100 task (see Fig. 8) [30]. Note that the momentum buffers were not reset between tasks; when they are reset, the momentum curve is nonmonotonic (see Fig. 9). Although momentum assists single-task performance, any amount of momentum will lead to forgetting previous knowledge.

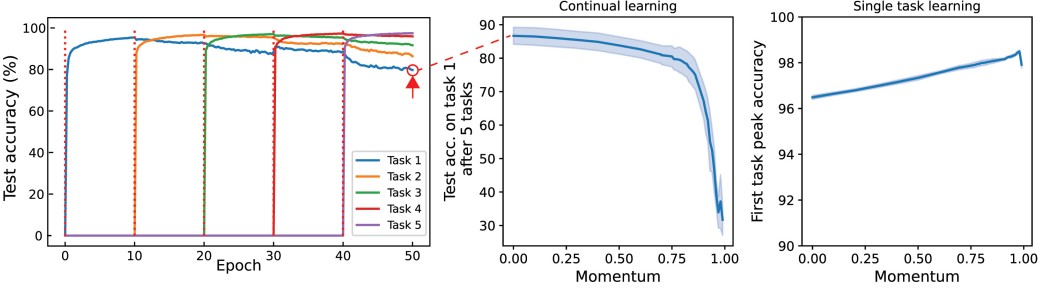

Figure 4: Effect of momentum in SGD on the Permuted MNIST task for an MLP with 2 layers and 1,000 hidden units. (middle) Test accuracy on the first task at the end of training 5 sequential tasks. (right) Final test accuracy on the first task before seeing the other tasks. Error bars represent standard deviations over seeds. See Appendix for further parameters.

## 5.2 Width improves remembering — but only for certain optimizers

As networks grow wider and (slowly) approach their infinite-width limit, they should remember better if one uses the appropriate posterior update rule over the Neural Tangent Ensemble.

Previous literature confirms that this is generally the case. In [40], the authors report the benefits of scale are robust across architectures, tasks, and pretraining strategies, although they largely use SGD with momentum $\beta = 0.9$. In [35], the authors report similar results and investigate other continual learning benchmark algorithms such as EWC ([22]). Forgetting seems to be largely solved by scale.

The reason for this in our framework differs from the reason cited by both [35, 40], which state that the gradients on different tasks will be more orthogonal in high dimensions, which reduces interference. Our interpretation is somewhat different and instead depends on the Jacobian of the network changes. We place no condition on gradient orthogonality between tasks. If the neural tangent experts are indeed fixed, the NTE update rule will find the multitask solution.

If this is the case, then wide networks should better remember only if the optimizer can be interpreted as a posterior update. In Fig. 5, we report that Adam's amnesia is not helped with increasing scale for the Permuted MNIST task. Although this could be for multiple reasons, we argue it stems from a divergence from a valid interpretation as a posterior update.

## 5.3 The NTE posterior update rule using current gradients improves with scale

In Section 4.2, we introduced a modified version of NTE posterior update rule in which the Jacobian at initialization replaced with the current Jacobian. As networks get wider, this algorithm will converges to the proper update rule due to the fact that the network Jacobian does not change in the lazy regime. This predicts that this rule will improve with scale. To test this, we trained an ML on Permuted MNIST and ConvNeXtTiny on task-incremental CIFAR100 with this approximate rule. We find that both single-task and multitask learning are greatly improved with width (Fig. 5 and Fig. 10). We take this as empirical evidence that the proper NTE posterior update (with a static Jacobian) would work well in the infinite-width limit.

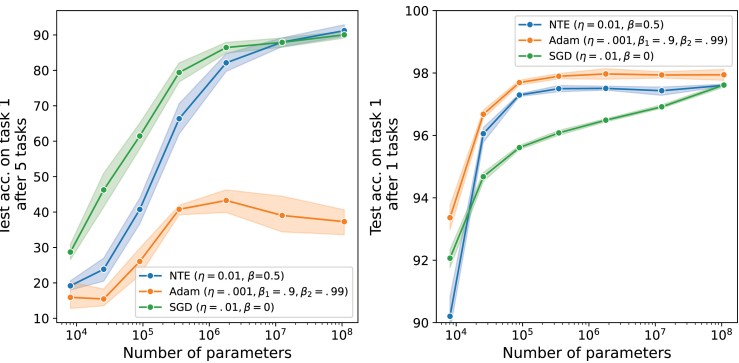

Figure 5: Wider networks forget less, unless trained with Adam. See Alg. 1. All networks are 2-layer MLPs with ReLU nonlinearities trained on 5 Permuted MNIST tasks. Loss curves and further parameters in Appendix. Error bars represent standard deviations.

## 6 Related work

There is a long history of interpreting networks as ensembles. Networks with dropout, for example, allow this interpretation [12, 14]. This is also closely related to Mixture of Experts models in classic [18, 20] and recent [43, 54] work. These approaches explicitly encode the experts within the network, and unlike our work do not use a Taylor expansion to establish the ensemble experts.

The idea of a Bayesian ensemble *over* networks is also well-studied. Such ensembles can either be assembled empirically through sampling [51, 50], built via a Laplace approximation [32], or optimized [3]. Bayesian posteriors are also common players in theoretical works using methods from statistical physics and PAC-Bayes [44, 27]. Some treatments of infinite-width limits study the ensemble of lazy learners [16]. While similar in spirit, these methods study groups of many networks rather than view a single network as an ensemble.

Finally, there is related work that uses ideas from ensembles for continual learning. Many of these are in the category of methods that continually learn by training new modules for each task

[49, 5, 42, 52, 39, 21]. Most directly related to this current work are papers that take a Bayesian approach and track statistics about the approximated posterior over networks [22, 10, 11, 28, 41, 37]. Many of these works in both categories require task boundaries. Furthermore, by introducing new modules or tracking statistics, these methods require additional memory to prevent forgetting.

## 6.1 Moving in directions of low curvature to forget less

Our framework justifies the strategy of encouraging parameters to change mostly in directions of low curvature. Such regularization methods are already well-established and proven to reduce forgetting [24, 31, 41]. Although not directly equivalent, this is also similar to Elastic Weight Consolidation, which penalizes by the Fisher Information matrix (an expected second-order derivative of the *log*-likelihood, rather than the likelihood) [22]. Another proximal method is Synaptic Intelligence, which penalizes parameter changes proportional to their integrated gradients along the path, which in the special case of diagonal, quadratic loss functions, is equivalent the Hessian [53]. Thus, a second interpretation of why these methods work well (and improve with scale [35]) is that they ensure the tangent experts in the NTE do not change much while learning.

## 7 Discussion

Here, we described how networks in the lazy regime can be seen as ensembles of fixed classifiers. With this perspective, we proposed weighing each expert by its posterior probability to form a Bayesian ensemble, and derived the update rule. This strategy of learning by posterior updating has the benefit that the order of data presentation does not matter – sequential experience and interleaved experience lead to identical ensembles.

The posterior update rule to the Tangent Ensemble is surprisingly similar to SGD on the model weights. However, it is interesting to note that this update rule is suboptimal. Posterior probabilities are the optimal ensemble weights only when the experts are independent [47, 34, 38] and well-specified [33, 6]. This assumption is violated by the use of shared data, as well as the fact that neural network architectures introduce dependencies between gradients. Although this does not affect the equivalence between the interleaved and sequential task performance (i.e. forgetting), this will reduce the performance of networks trained with the NTE posterior update. This suggests avenues for improving SGD.

This suboptimality could be addressed in multiple ways. In the ensemble literature, there are many strategies to diversify the expert pool [4] such as repulsion [9]. Different experts might also be trained on different data [46], and one might even take a boosting approach [15]. It is very likely that these approaches would yield neural networks that outperform standard networks trained by updating the posterior distribution over tangent experts.

The ability in interpret single networks as ensembles opens many avenues for future research. These extend beyond continual learning; for example, one might be able to obtain a measure of uncertainty of the network output via the variance of the experts [12]. We are hopeful that this insight will lead to deep learning systems that are better understood as their use expands within society.

## Acknowledgements

The authors thank Peter Koo and Ben Cowley for helpful early conversations and Tony Zador for providing a collaborative research environment. Additionally we would like to thank a grant from Schmidt Futures to CSHL for funding.

## Code availability

The code for all figures in this paper were written in Jax and are available at https://github.com/ZadorLaboratory/NeuralTangentEnsemble.

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

# 8 Appendix

## 8.1 Proof of Theorem 2

*Proof.* We begin by noting that the change in weights we can be split up the sign and magnitude,

$$\Delta w_i = |\Delta w_i|\text{sign}(\Delta w_i).$$

We will then interpret $|\Delta w_i|$ as the unnormalized component weight. The remaining terms must be the component functions.

To identify these functions, and show that they satisfy the properties of a probability distribution, we will rearrange terms. First, noting that $\sum_i |\Delta w_i| = z$ for some constant $z$ (potentially $z = 1$),

$$p(y|x, W^{(t)}) = p(y|x, W^{(0)}) + \sum_i^N |\Delta w_i|\text{sign}(\Delta w_i)\frac{\partial}{\partial w_i^{(0)}}p(y|x, W^{(0)}) + \mathcal{O}(\|\Delta W\|^2) \quad (17)$$

$$= \sum_i^N \frac{|\Delta w_i|}{z}\underbrace{\left(p(y|x, W^{(0)}) + z\,\text{sign}(\Delta w_i)\frac{\partial}{\partial w_i^{(0)}}p(y|x, W^{(0)})\right)}_{p(y|x,f_i)} + \mathcal{O}(\|\Delta W\|^2)$$

$$\quad (18)$$

$$= \sum_i^N \frac{|\Delta w_i|}{z}p(y|x, W^{(0)})\left(1 + z\,\text{sign}(\Delta w_i)\frac{\partial}{\partial w_i^{(0)}}\log p(y|x, W^{(0)})\right) + \mathcal{O}(\|\Delta W\|^2)$$

$$\quad (19)$$

We call the term $p(y|x, f_i)$ the ***neural tangent expert***.

The neural tangent expert provides a valid probability distribution for small $\Delta W$. First, see that it satisfies $\sum_j p(y_j|x, f_i) = 1$. This can be seen from the fact that the right term inside $p(y|x, f_i)$ (the parentheses in the middle line) sums to 0 over the output label:

$$\sum_j p(y_j|x, f_i) = \sum_j z\,\text{sign}(\Delta w_i)\frac{\partial p(y_j|x, W^{(0)})}{\partial w_i^{(0)}}$$

$$= z\,\text{sign}(\Delta w_i)\sum_j p(y_j|x, W^{(0)})\frac{\partial \log p(y_j|x, W^{(0)})}{\partial w_i^{(0)}}$$

$$= 0$$

Here we have used the identity that the expectation of a score function is zero.

Next, we will show that $1 \geq p(y_j|x, f_i) \geq 0$. First, since each $p(y_j|x, f_i)$ sum to 1 over $j$, no component can be greater than 1 if all components are nonnegative. Thus, it only needs to be shown that $p(y|x, f_i) \geq 0$. While this cannot be guaranteed in general, by construction we have assumed that $zL < 1$. This Lipschitz continuity bounds the L2 norm of the gradients of the log likelihood, which in turn bounds the L1 norm and implies that any *individual* gradient has magnitude less than $\frac{1}{z}$:

$$z\left|\frac{\partial \log p(y_j|x, W^{(0)})}{\partial w_i^{(0)}}\right| < 1$$

Thus, whether $\text{sign}(\Delta w_i) = 1$ or $\text{sign}(\Delta w_i) = -1$, the term in parenthesis is nonnegative.

$$\left(1 + z\,\text{sign}(\Delta w_i)\frac{\partial}{\partial w_i^{(0)}}\log p(y|x, W^{(0)})\right) > 0.$$

$\square$

## 8.2 Preventing component functions from changing by keeping the network close to initialization

The continual learning ability of a Bayesian ensemble derives from learning to weight a set of *fixed* functions. If these functions change over time, then there is no guarantee that the likelihood of each function at time $t$ still reflects the cumulative likelihood of past data under the current function.

One good way to ensure this is does not occur is to ensure that the parameters change as little as possible from initialization. Although it is typical to measure this distance with $\|\Delta W\|_2$, we instead measure distance as the relative entropy of the change in parameters from the uniform perturbation, due to the simplicity of its result. These have the same minimum; remembering that $\|\Delta W\|_1 = 1$, by normalization, the smallest Euclidean distance $\|\Delta W\|_2$ will occur when all $\Delta w_i$ are equal.

To derive the maximum-entropy vector $|\Delta W|$ that is as similar as possible to $p(f_i|\mathcal{D})$, we will follow the steps of [23]. A first step is to set the notion of similarity $L$ between $|\Delta W|$ and $p(f_i|\mathcal{D})$. We will then find the value that minimizes:

$$U(|\Delta W|) = -H[|\Delta W|] + \beta L(|\Delta W|, \{p(f_i|\mathcal{D})\}) + \gamma(\|\Delta W\|_1 - 1)$$

Here $\beta$ is a parameter that trades off between entropy and matching $p(f_i|\mathcal{D})$, and $\gamma$ is a Langrange multiplier that ensures the parameters remain normalized.

If one chooses to maximize the dot product $|\Delta W|^T p(f_i|\mathcal{D})$, one obtains the following relation:

$$w_i = \frac{e^{\beta\,p(f_i|\mathcal{D})}}{\sum_i e^{\beta\,p(f_i|\mathcal{D})}}$$

Alternatively, if one chooses to minimize the relative entropy $\mathrm{KL}(|\Delta W|, p(f_i|\mathcal{D}))$, then one obtains

$$w_i = \frac{p(f_i|\mathcal{D})^\beta}{\sum_i p(f_i|\mathcal{D})^\beta}$$

We implement this second term. If the posterior likelihoods are maintained in log space, $\beta$ acts as a multiplicative scale upon the log data likelihood.

## 8.3 Pseudocode for the NTE update rule using current gradients

---
**Algorithm 1** Neural Tangent Ensemble posterior update rule with current gradients

---
Receive a dataset $\mathcal{D} = \{x_k, y_k\}_{k=1}^{N_t}$
Initialize a neural network with parameters $W^{(0)}$
Set learning rate $\eta$ and discount factor $0 < \beta \leq 1$
Perturb the network with some $\Delta W$ such that $\|\Delta W\|_1 = z$
**for** each example $(x_k, y_k) \in \mathcal{D}$ **do**
    **for** each edge $w_i \in W$ **do**

        Compute the data likelihood for each expert $p(y_k|x_k, f_i) = \left(1 - \eta\,\mathrm{sign}(\Delta w_i)\frac{\partial}{\partial w_i^{(t)}}\ell_k^{(t)}\right)$

        Update perturbation multiplicatively $\Delta w_i \leftarrow \Delta w_i\, p(y_k|x_k, f_i)^\beta$
    **end for**
    Renormalize perturbation such that $\sum_i |\Delta w_i| = z$
    Optionally clip the change in parameters to prevent large changes, such that $|\Delta w_i| = 1$
**end for**

---

## 8.4 Experiment details

All MNIST experiments were completed on two NVIDIA RTX 6000 cards, and all CIFAR100 experiments were conducted on NVIDIA H100 cards. Over 1,500 individual models were trained across all seeds and conditions, amounting to roughly 440 GPU-hours of compute time.

### 8.4.1 Figure 2

A single MLP was trained with 1,000 hidden units per layer and 2 hidden layers using ReLU nonlinearities. The model perturbed from initialization with a random normal vector with scale 0.001, and then was trained with the NTE update rule (Algorithm 1) but using the Jacobian of the model at initialization. The batch size was 24 and the parameters of the NTE algorithm were $\eta = 0.01$ and $\beta = 1$.

### 8.4.2 Figure 3

We first created a Permuted MNIST task and code to measure the test accuracy on all tasks after training on each task sequentially. All reported results use 5 tasks.

We trained an MLP on this task with ReLU nonlinearities and 1,000 hidden units in 2 hidden layers. We used SGD with batch size 128, learning rate 0.01, and momentum swept from 0 to 1. The momentum buffer was not reset between tasks. We report the standard deviation of 10 random seeds.

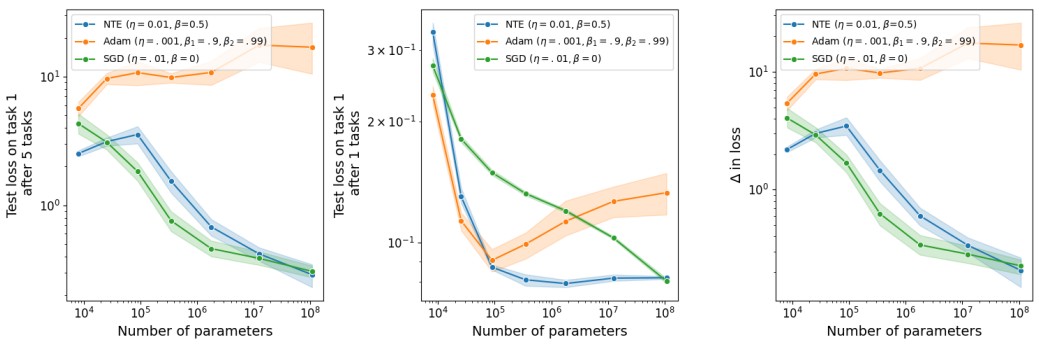

Figure 6: Loss curves for the task in Fig. 4.

### 8.4.3 Figure 4

Here, we used the same continual learning task as Figure 3, but swept the width of the two hidden layers from 10 to 10,000. All batch sizes were 128.

## 8.5 Additional experiments

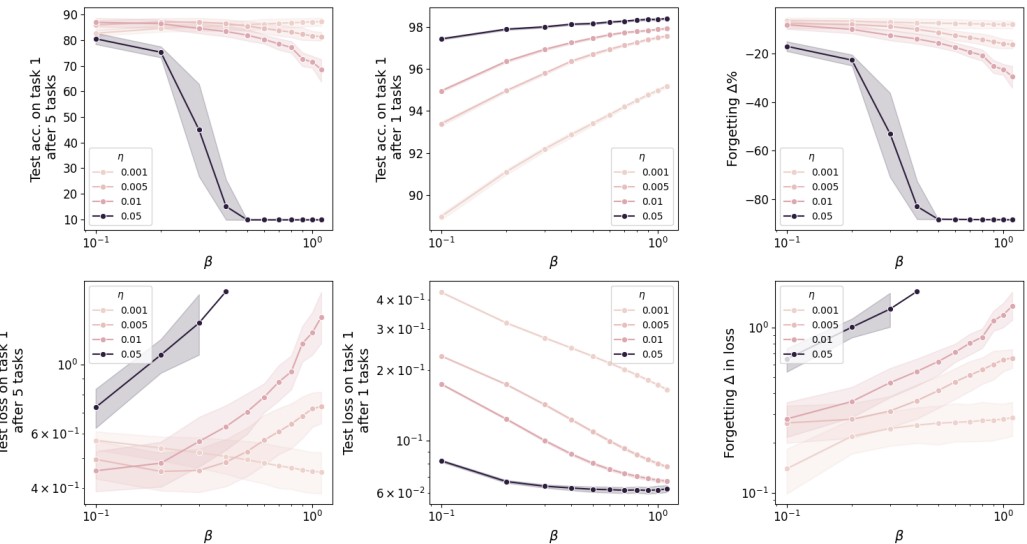

Figure 7: For the same task and architecture as the other figures (Permuted MNIST for 5 tasks with a ReLU MLP with two hidden layers and 1,000 hidden units each), we swept the parameters $\beta$ and $\eta$ in the Algorithm above. The accuracies (top) and losses (bottom) are shown for the first task after 5 total tasks (left), after immediately finishing that task before task switching (middle) and the difference between the two (right). Error bars show the standard deviation across seeds.

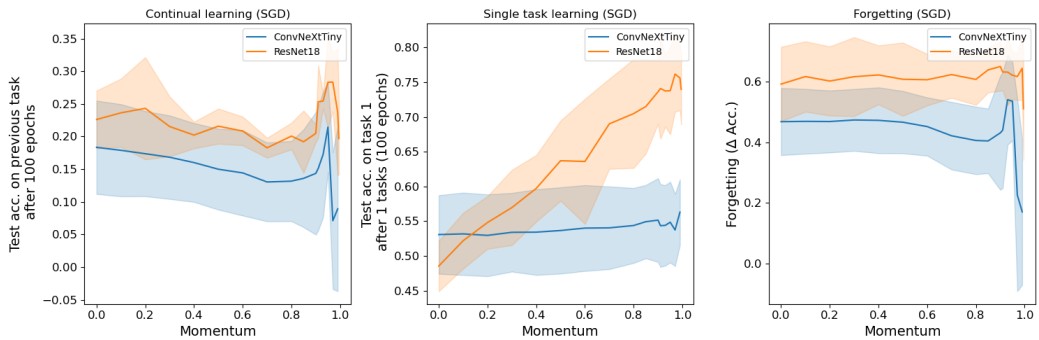

Figure 8: Effect of momentum in SGD for modern CNN architectures trained on the CIFAR-100 task-incremental task. In this task, models are trained on 10/100 classes at a time, and the softmax output layer is masked to only the active classes. Each model is trained for 100 epochs per task, and evaluated on all previous tasks. The two models shown are a ResNet18 and a ConvNeXtTiny. (left) The test set accuracy on the immediately previous task after learning the final task. (middle). The test set accuracy on the first task. (right) The difference between the two plots to the left.

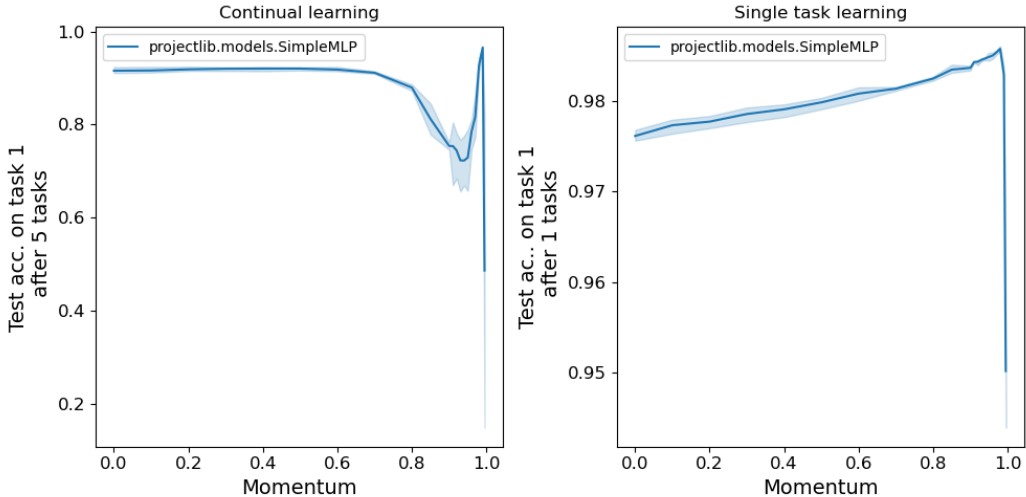

Figure 9: Identically to Fig. 4, we trained a 2L MLP with 1,000 hidden units on the Permuted MNIST task and varied the momentum of SGD. This time, we reset the momemtum buffer between tasks. Interestingly, this introduces a nonmonotonic behavior and one can attain good performance with momentum near 0.99.

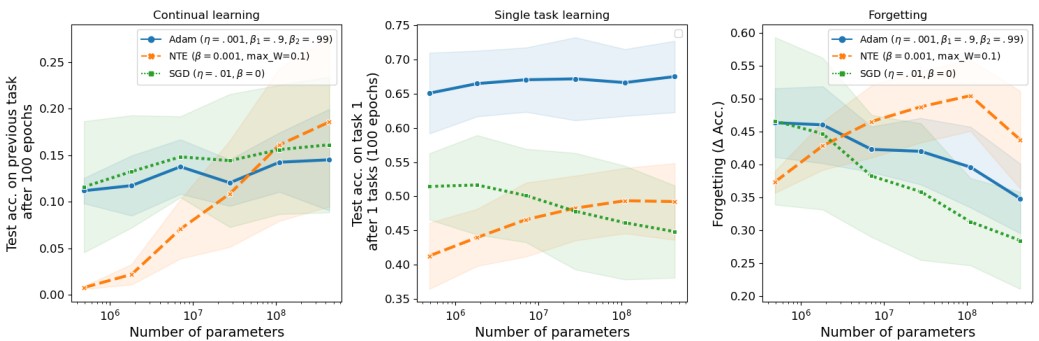

Figure 10: Performance with width for CIFAR-100. We scaled the number of convolutional filters in all layers of a ConvNeXtTiny by a constant factor, and then trained on the CIFAR-100 task-incremental task for each network. Subpanels represent identical information as Fig. 8.

