# OpenReview forum: "Continual learning with the neural tangent ensemble"
_NeurIPS.cc/2024/Conference — NeurIPS 2024 spotlight_

### Official Review · Reviewer_Yq9r · 2024-07-01

**Soundness:** 3
**Presentation:** 3
**Contribution:** 3
**Rating:** 7
**Confidence:** 3

**Summary:**

This is a theoretical paper that seeks to find an algorithm to train neural networks without forgetting in continual learning settings.

The authors show that in the infinite width limit, a single classification network can be reformulated as a weighted ensemble of fixed classifiers (fixed experts).

They provide a learning rule, which computes the posterior distribution of each expert.

This learning rule turns out to be similar to SGD, where there is a per-parameter learning rate and the different of the parameters to their initial state is constrained to be L1 normalized (unsure of the exact correct phrasing of the L1 projection step).

In the finite case, the experts can no longer be viewed as static.

Experiments show that any amount of momentum hurts continual learning performance, but momentum does help when learning a single task from scratch.

**Strengths:**

Theoretical understanding of how to prevent forgetting in networks as training continues is important and helps move the field forward.

The connection of SGD to posterior updates is interesting and (as far as I'm aware) novel.

The experiments exploring the effect of momentum on a networks ability to continually learn are interesting and useful from a practitioner's perspective.

Experiments demonstrate that the NTE update rule becomes more appropriate as width increases.

The work opens up future directions that may be able to estimate uncertainty based on the variance of experts under the NTE formulation.

**Weaknesses:**

The claim that Bayesian ensembles of fixed functions are natural continual learners does not seem obvious to me. In the infinite-width I buy it, but for finite networks, it seems like it would be possible to prune all of the experts away leaving you with a degenerate model.

Only one continual learning benchmark dataset (permuted MNIST) is used in the experiments. The details of this task are not well explained.

The introduction highlights the value of Bayesian posterior updates as being order-independent. However, there is no experiment exploring the sensitivity of the NTE update rule to the ordering of the data.

Section 5.4 seems out of place. It isn't a result, and it seems a stretch to call it a prediction. It may be useful reducing it to a few sentences in the related work section so other sections can include more clarifying text.

Overall, the proofs seem correct, but that required a significant amount of time and further reading for me to come to that conclusion. Such theoretical work would be much stronger if it codified its proofs in the form of a proof assistant like Lean4 or Coq. This would require the reviewer to only check that the claim is correctly formulated. The proof itself would be verified automatically, and there would be no question about its correctness.  Codified proofs would allow readers learn more about any unfamiliar identities used with full confidence and remove all ambiguity stemming from unfamiliar notation.

Typo: Line 205. "This is constrains" should be "This constrains"

**Questions:**

What happens if there is label noise? This analysis seems to assume that the dataset is perfect, which raises the question: if you have an idealized network that cannot forget, what is the impact of a mislabeled example?

The paper claims that in the infinite-width network each parameter/weight/edge contributes a classifier to the ensemble. This seems to imply that weights in earlier layers are included here, is that true? I'm confused about how to picture this. In the first layer, there is a weight connecting one of the finite input neurons to one of the infinite hidden neurons. That particular edge isn't able to see much information, but its output does connect to an infinite number of neurons in the second hidden layer. Is its interpretation as a weight on the neural-tangent-experts somehow implicit wrt to the rest of the network depth? Is it that the edge allows energy to accumulate in the "right spots"?

Similarly, in the context of finite networks, is there a way to extract the individual experts? Are the  number of experts still equal to the number of model parameters in this case?


Figure 3: I'm a confused about the continual learning task setup for permuted MNIST. It would be useful to remind readers what the mechanism for predicting on task 1 after 5 tasks is. Is there a new head? Do the number of output units stay fixed throughout all tasks?

In theorem 1: why is the perturbation important?  Is ΔW the linear update that brings an untrained NTK weights (W⁰) to the trained NKT weights (Wᵗ)? If this is true, it might help to the statement of theorem 1 to call that out.

Can anything be said theoretically about how fast increasing the width of a network converges towards the NTE limit?

In Figure 4 are the curves over just the first task? In other words, is this just training on regular MNIST to show how the NTE update rule begins to fail as the weights drift from their initial state?

**Limitations:**

The authors are clear about the technical limitations of the work, and do not
make any unreasonable claims.

Code to reproduce looks like it exists.

---

> ### Author Rebuttal · Authors · 2024-08-07
>
> >The claim that Bayesian ensembles of fixed functions are natural continual learners does not seem obvious to me....
>
> We have added a Lemma that describes more precisely why weighing fixed functions according to their posterior probability is a learning strategy which does not depend on the order of the data.
>
> To your second point, since the weights reflect the posterior probability, they must sum to 1. This makes it impossible to prune away all of the experts. However, there does exist a degenerate case where only 1 expert remains, yet even in this case the weight $w=1$ on this model is invariant to data ordering
>
> >Only one continual learning benchmark dataset (permuted MNIST) is used in the experiments...
>
> We have added an additional benchmark of CIFAR-100 in the task-incremental setting in which only 10 output classes are observed at a given time. We will add further experimental details to the revised manuscript.
>
> >The introduction highlights the value of Bayesian posterior updates as being order-independent...
>
> We hope that the Lemma above is sufficient to clarify this point. This is a guarantee of the fact that the Bayesian posterior is obtained via the likelihood of the data under each expert, which depends on data likelihood multiplicatively.
>
> >Section 5.4 seems out of place...
>
> We agree and will move this text.
>
> >Overall, the proofs seem correct.... Such theoretical work would be much stronger if it codified its proofs in the form of a proof assistant like Lean4 or Coq...
>
> Thank you for this detailed reading! For the current manuscript, we regret that we did not have sufficient time in the rebuttal period to codify this work in a proof assistant.
>
> >Q1: What happens if there is label noise? ...
>
> A1: This is an important point. This is a drawback of this problem setting: the goal of being invariant to the order of data inherently treats all datapoints equally. We note that this is a general feature of this domain, also a problem of full-dataset Gradient Descent, and that it is standard to train networks by the likelihood of the dataset under the model. However, it is an interesting line of research to combine robustified learning algorithms with continual learning.
>
> >Q2: ... This seems to imply that weights in earlier layers are included here, is that true? ...
>
> A2: Indeed, each edge contributes an expert – even the lower layers. This is true both for finite networks and infinite-width networks. This is indeed surprising – as at first glance it seems odd that input layers should be contributing a probability distribution over the output classes.
>
> We find it helpful to imagine gradient flow. Take some edge E in the middle of the network. There are many input/output paths that traverse edge E as they ascend from the inputs to the outputs. The weight of edge E acts as a gain control on all of these input/output paths. When we slightly perturb edge E, the effect upon the output classes due the sum of all of these paths slightly changes as well. This is how the architecture of the remaining network enters: the effect of edge E depends on gradient flow through all upstream and downstream edges. It is this perturbative effect upon the output classes that gives edge E its additive effect upon the output logits. Theorem 1 then states further that these perturbations themselves act as a valid classifier, thus allowing us to interpret the entire network as an ensemble over the perturbative effects of each edge in the entire network.
>
> We can state this slightly more formally. The effect of a single edge (expert) on the output of the network is given w.r.t. the other edges across all layers in the network. Specifically, in the NTK, the Jacobian controls how each edge affects the output. Consider a single entry of the Jacobian, $df_i / dw_j$, which controls how the perturbation around edge $w_j$ adjusts a single output of the network. This quantity is the gradient path from $f_i$ to $w_j$. For finite networks, a single column of the Jacobian can be understood as an expert, and the number of experts (length of the column) is equal to the number of parameters.
>
> >Q3: Figure 3: I'm a confused about the continual learning task setup for permuted MNIST....
>
> A3: For permuted MNIST, nothing changes about the network from task to task since the output labels are always the same. Only the pixels in the input are permuted.
>
> Additionally, we will clarify our continual learning setups by using the vocabulary of van de Ven et al. 2022 (https://www.nature.com/articles/s42256-022-00568-3). The Permuted MNIST task is an example of domain-incremental learning. In contrast, the CIFAR-100 task which shows 10 classes at a time is an example of task-incremental learning.
>
> >Q4: In theorem 1: why is the perturbation important? ...
>
> A4: Yes, $ΔW$ is the change from the untrained weights to the trained weights. It is important for Theorem 1, because the magnitude of the perturbation is the quantity that we define as the posterior probabilities (i.e. weights on the experts).
>
> In general, the perturbation is key due to our reliance on a first-order Taylor expansion describing the network at $f(W+∆W)$. Coarsely, in order for a network with N edges to be a Mixture of Experts with N experts, there needs to appear a $\sum_i^N$.
>
> >Q5: Can anything be said theoretically about how fast increasing the width of a network converges towards the NTE limit?
>
> A5: This is a complex question. We are aware of some work describing how quickly networks approach the lazy regime, but only in certain settings. We have attached an empirical study of how quickly the NTE limit is achieved with width for MLPs on the MNIST task.
>
> >Q6: In Figure 4 are the curves over just the first task? ...
>
> A6: Might you mean Figure 2? For Figure 2, this a network just trained on regular MNIST to show the effect that you describe, and yes, this is just the first task it has seen.

---

> > ### Comment · Reviewer_Yq9r · 2024-08-12
> >
> > Thank you for the response this clarifies much of what I was unclear on.
> >
> > > For finite networks, a single column of the Jacobian can be understood as an expert, and the number of experts (length of the column) is equal to the number of parameters.
> >
> > This clears things up for me and may be worth stating in the paper itself.
> >
> > There is an additional question I have with respect to the momentum experiment: when using momentum, was the optimizer state reset between task changes? I think the experiment is interesting regardless of this, but I think this is an important detail to state.
> >
> > Overall I think this is a strong paper and recommend accepting.

---

### Official Review · Reviewer_q1Zy · 2024-07-07

**Soundness:** 3
**Presentation:** 2
**Contribution:** 3
**Rating:** 5
**Confidence:** 2

**Summary:**

This is a theoretical research on preventing forgetting by considering a single network trained with a lazy-regime as an ensemble model of multiple functions and adjusting their weights.

**Strengths:**

The discussion is based on solid theory. This paper obtained an insight that the posterior update rule for the NTE is equivalent to a scaled and projected form of stochastic gradient descent without momentum. Because of this relationship, this paper successfully demonstrates the disadvantages of using momentum and the advantages of increasing width which may induce lazy training.

**Weaknesses:**

I feel that the contribution is valuable, but some settings might be difficult to realize in the context of NTK, even with an infinite limit. I might be able to deepen my understanding through the answers to the questions, so please refer to the Questions section.

**Questions:**

**(1) Non-linear output**

Many discussions are based on the formulation of Equation (5), and I assume this setting is intended to be used with non-linear transformation such as softmax. However, in my understanding, the NTK does not remain constant in such cases [1]. What are your thoughts on this perspective? If I have misunderstood, please correct me.

**(2) NTK regime**

There is a lack of discussion regarding the settings of the model used in this study. In order for the NTK to remain constant during training, parameter scaling or output scaling must be set appropriately. In some settings, lazy training would not be realized, even if the width is increased [2]. In the experiments, does the NTK indeed stop changing as the width increases?

**(3) Suboptimality**

As mentioned in the Discussion section, experts are not actually independent, so the weighting strategy discussed is not actually optimal. In the Discussion section, various ideas are presented for training experts to maintain independence, but these approaches have not been evaluated in this paper. Therefore, it seems important to show that the impact of suboptimality is not significant. Is that possible? If the impact is too large, I feel that the premise of the discussion might be a bit fragile.


[1] Liu et al., On the linearity of large non-linear models: when and why the tangent kernel is constant, NeurIPS 2020.

[2] Yang and Hu, Feature Learning in Infinite-Width Neural Networks, ICML2021

**Limitations:**

Discussion is only effective within a lazy regime. Additionally, experts are required to be independent.

---

> ### Author Rebuttal · Authors · 2024-08-07
>
> For Q1 and Q2, there are really two possible statements in question. The first is whether a particular network at any single moment in time can be seen as an ensemble of valid classifiers. Theorem 1 describes how this is always the case as long as the network is Lipschitz, as one could construct a seed point for the Taylor expansion arbitrarily close to the network. This result stands alone, and could be used in further work to design quantifications of uncertainty, etc.
>
> The second question is whether and when these classifiers evolve in time due to optimization. We agree that we did not adequately expand on this question. In revision, we will include a discussion of this issue and include our responses to these questions.
>
> > Q1: Many discussions are based on the formulation of Equation (5), and I assume this setting is intended to be used with non-linear transformation such as softmax. However, in my understanding, the NTK does not remain constant in such cases [1].
>
> Yes, in our setting we deal with softmax nonlinearities. Thank you for this reference. We agree that the presence of this output nonlinearity adds structure to the Hessian which does not diminish with width resulting in changes to the NTK kernel for standard initializations. However, we want to emphasize that this does not affect Theorem 1 in itself, which again, relates to whether a linearized network with an arbitrary ∆W can be interpreted as an ensemble. This only relies on the Lipschitz continuity of the network, and one can always choose a perturbation ∆W arbitrarily close to the initial point such that the linearization remains valid. Thus, this question is about how much the tangent experts change - not whether they are indeed interpretable as experts.
>
> It is a second matter whether the ∆W given by a specified learning algorithm, data, and objective will keep the Jacobian fixed. The results from [1] do indeed suggest that nonlinearities will make this more difficult and perhaps not be achievable even in the infinite width limit. (As an aside, even for softmax nonlinearities a truly lazy regime can still be achieved by scaling the outputs as in Chizet et al, which the authors of [1] mention in their Appendix A.) Even when experts change, minimizing this change remains a goal to reduce forgetting. This can be achieved by making networks wider, but also through other strategies such as the nonlinearities or moving preferentially in directions of low curvature.
>
> In response to Q1 and Q2, we empirically evaluated whether width scales how much experts change under the NTE update rule. We trained several MLPs of increasing width on MNIST. For each expert, we then calculated how much it changed from initialization via the squared difference of its respective column in the Jacobian from initialization. (Each entry in an expert’s column relates to its effect upon an output probability). We then reported the average of this distance over all experts, and this plot is attached in the pdf. We found that increasing width leads to a diminishing change in the experts, meaning that wider networks will indeed forget less.
>
> > Q2: There is a lack of discussion regarding the settings of the model used in this study. In order for the NTK to remain constant during training, parameter scaling or output scaling must be set appropriately. In some settings, lazy training would not be realized, even if the width is increased [2]. In the experiments, does the NTK indeed stop changing as the width increases?
>
> We will add a discussion of this important issue. This relates to our paper's central message: keeping the NTK fixed is crucial for continual learning. Thus, in practice, every effort should be made to ensure the network learns lazily. In principle, this could even be used as the basis for the design of new initializations that forget less.
>
> A related possibility is that one might try to derive a rule to weigh the ensemble while remaining in the NTK regime. This motivates the use of the regularizer $\eta$, which is currently derived in Appendix 8.2. In Section 4.1 and 4.2, we then motivate a rule where the Jacobian around the current weights is used for linearization. This matches the reasoning in [2] that a network must not only weigh the experts (features) at initialization, but also learn useful experts (features) for the task at hand. Furthermore, in Section 5.4, we explain that moving along low curvature directions allows the network to stay in a linear regime. This aligns with the results of [1] where a transition to NTK constancy is given by a Hessian with shrinking spectral norm. While we cannot guarantee that all directions of the loss landscape become shallow with increasing width, we can choose only to move in the directions that do, effectively matching the settings of [1] in practice. Implementing a practical rule under this guidance is out of the scope of our paper and left for future work.
>
> >Q3: As mentioned in the Discussion section, experts are not actually independent, so the weighting strategy discussed is not actually optimal...
>
> A3: The recognition that this weighing strategy is suboptimal was an interesting finding to us – especially given our finding that the posterior update rule is remarkably similar to SGD. Thus, due to our derivation, we are able to see that this SGD-like rule is in fact suboptimal. We think it would be an interesting line of future work to actually improve upon SGD by taking the ensemble interpretation of wide NNs and penalizing experts for their redundancy.
>
> For the time being, however, our empirical results show that this suboptimality does not prohibit taking this overall approach. We are able to learn tasks with matched performance with standard optimization while maintaining the ability to continually learn. Furthermore, we would like to highlight again that the posterior update rule for the NTE is, surprisingly (to us), very similar to SGD, which in practice of course works quite well.

---

> > ### Comment · Reviewer_q1Zy · 2024-08-08
> >
> > Thank you for your response. My concerns have been addressed by the additional experiments. Therefore, I will update my score from 4 to 5.

---

### Official Review · Reviewer_MP6D · 2024-07-12

**Soundness:** 3
**Presentation:** 3
**Contribution:** 2
**Rating:** 6
**Confidence:** 2

**Summary:**

The paper introduces a novel approach to mitigating catastrophic forgetting in neural networks by introducing the concept of Neural Tangent Ensemble (NTE), a formulation interpreting a single neural network as an ensemble of fixed classifiers, leveraging the Neural Tangent Kernel (NTK) framework. This interpretation allows for the derivation of a Bayesian posterior updating rule, which is shown to be equivalent to a scaled and projected form of stochastic gradient descent (SGD). The paper presents theoretical insights into how neural networks can be viewed as Bayesian ensembles, offering an optimization-guided framework for understanding and addressing forgetting in continual learning scenarios.

**Strengths:**

1. Theoretical rigor: The authors provide a detailed theoretical analysis, including proofs and derivations.
2. Connection to SGD: The finding that the NTE posterior update rule is connected to SGD is insightful and could have implications for optimization strategies in neural networks
3. Clear presentation: The paper is clearly presented with fluent logic.

**Weaknesses:**

To me, the largest limitation of this paper is the empirical validation. Although the paper includes empirical results that support the theoretical claims, the scope of experiments could be expanded. The experiment is implemented on a relatively simple dataset (Permunated-MNIST). Additional experiments on more complex datasets would be appreciated.

Beyond that, the comparison is relatively insufficient. Could the authors provide comparisons with other optimization-based continual learning methods, like OGD or others, to validate the effectiveness of the proposed method? Or are there any other reasons that they can not be directly compared?

**Questions:**

1. Are there any additional experiments that could be conducted to further validate the effectiveness of the NTE approach in more complex or real-world continual learning tasks?
2. How does the performance of NFE compare with other optimization-based methods?
3. Could the author elaborate more on Equation(3) and line 78, "It is easy to create multitask Bayesian ensembles even when tasks are seen sequentially," which is not very direct to me?

**Limitations:**

The authors have outlined the limitations and the potential societal impact of the work.

---

> ### Author Rebuttal · Authors · 2024-08-07
>
> Thank you for appreciating the theoretical novelty and generality of the NTE idea. Following these suggestions, we have implemented several more empirical characterizations with modern CNN architectures and the more complicated CIFAR-100 incremental learning task.
>
> We would like to emphasize that the contributions of our theoretical work can be somewhat separated from the experiments. Our theory provides a rigorous statement about infinite-width networks, which as a novel bridge between ensembles and NNs should be of general interest. Like many theoretical observations, including the NTK paper, this is not directly applicable but instead can be approximated. The approximation we evaluate – which uses the current gradients instead of the gradients at initialization – is designed as an exploration of this limit. Thus, we view this paper more as a new framework for neural network interpretation and a discussion of under what limits networks forget, and less as a paper documenting a new SOTA continual learning algorithm.
>
> >Q1: Are there any additional experiments that could be conducted to further validate the effectiveness of the NTE approach in more complex or real-world continual learning tasks?
>
> A1: Yes, we have expanded our empirical evaluation to include additional continual learning tasks as well as complex, real-world networks. We have now re-evaluated our main results on CIFAR-100 in the task-incremental setting, in which 10 classes are seen at a time, and using modern architectures such as ResNet and ConvNeXt.
>
> >Q2: How does the performance of [NTE] compare with other optimization-based methods?
>
> A2: In Section 3.2, we provide an insightful and surprising result that our NTE-based belief update closely matches SGD updates (with some constraint). So, we should expect that for single task learning, our rule performs just as well as standard SGD training. As such, this will underperform known methods for continual learning for certain networks (e.g. small ones with high curvature), such as EWC. We do not believe that this needs to be shown visually, as it should be clear from the theoretical result.
>
> Our result is instead designed to illustrate how describe how SGD will perform optimally at continual learning in the lazy learning limit. This was not obvious to us before our derivation. As an aside, we wish to emphasize our framework can be used to justify existing SOTA continual learning methods. EWC, for example, moves in directions of low curvature, exactly as needed to keep the tangent experts fixed.
>
> >Q3: Could the author elaborate more on Equation(3) and line 78, "It is easy to create multitask Bayesian ensembles even when tasks are seen sequentially," which is not very direct to me?
>
> A3: This fact is important to understanding the continual learning contribution of our paper. We start from a theoretical framework for describing networks as ensembles of fixed functions, then use Bayesian ensembling to address the continual learning problem. Referring to Fig. 1, given several tasks (e.g. Task A and Task B) and a set of experts, ${f_i}$, different subsets of experts will solve each task well. We seek to find the intersection of these subsets to identify the experts that solve all tasks sufficiently well. As the Lemma in the general comment states, the order in which tasks arrive sequentially is the order of the multiplication of weightings. Consequently, since multiplication is commutative, it does not matter which order we perform the multiplication; we will still arrive at the same final posterior weighting. Given the importance of this argument to our work, we will clarify the logic in the section that you have highlighted by walking through an example with multiple tasks.

---

> > ### Comment · Reviewer_MP6D · 2024-08-10
> > **Further Question**
> >
> > It seems that in more complex scenarios in CIFAR-100, the proposed NFE behaves much poorer than ADAM and SGD in small parameter sizes. Could the authors kindly elaborate on the possible reasons, especially the comparison between SGD and NEF, as long as neither introduces momentum?

---

> > > ### Author Response · Authors · 2024-08-12
> > >
> > > What we intent to show with this plot is that the NTE rule improves with network width, which it does - and note that for very large networks, it actually outperforms SGD. Underperformance is only true for smaller networks. This is to be expected, as the Neural Tangent Ensemble rule is derived from an infinite width limit. We think this is more pronounced for this architecture because for ConvNets the relevant NTK limit is over the number of filters per layer, rather than the overall number of parameters per layer. In the plot above, the number of filters ranges from 12 to 1028. Thus we are displaying more of the regime father away from the NTK limit, yet as we approach it, NTE rapidly improves.

---

> > > > ### Comment · Reviewer_MP6D · 2024-08-13
> > > >
> > > > Thank you for your explanation. I appreciate the additional experiments and I will raise my score.

---

### Official Review · Reviewer_Gy8g · 2024-07-15

**Soundness:** 4
**Presentation:** 4
**Contribution:** 4
**Rating:** 6
**Confidence:** 2

**Summary:**

The paper suggested that linearized networks (under lazy learning regime) at the initialization can be understood as an ensemble model where each ensemble component is a function parameterized by a single parameter in the network (i.e. if the network has N parameters then the prediction is an ensemble over N functions). The weight of each ensemble is controlled by the posterior weights of each ensemble/expert conditioned on the data in a particular task. Under this regime, when a new task comes in, experts good at previous tasks but bad at this task would be down-weight, as such the ensemble would continuously be adapted to new tasks without losing ability in previous tasks. Interestingly, the paper shows that the posterior update formula is almost identical to SGD update (without momentum). Additionally, the authors proposed a more practical version of the framework under the rich regime, where each ensemble/expert is evolving throughout time, under the assumption that the parameter does not move too far away from the initialization. Lastly, the authors draw three useful insights from the ensemble interpretation, and empirically verifies the insights on 2 layer NN on permuted MNIST.

Overall I find the connection between linearized network, ensemble and continual learning very interesting and novel. The theory is overall sound to me and the empirical results support the theory well.

**Strengths:**

- The interpretation proposed is novel and interesting.

- The posterior update perspective guarantees that under the proposed NTE theory, the model after seeing multiple tasks sequentially is identical to the one that sees all tasks jointly.

- The problem studied (catastrophic forgetting) is of great importance.

- The theory is mostly sound and the assumptions are reasonable.

- The application of the interpretation gives useful insights, such as momentum causes forgetting.

**Weaknesses:**

- It is unclear to me, under the Neural Tangent Ensemble theory, how far can the model change from the initialization, if the model is not changing too much, then it is not surprising that the model is not forgetting, since it is not learning a lot information from the training data.

- Similarly, the theory suggest that the model forgets less when getting wider, but the representation learning flexibility would also drop. So the model is learning less while forgetting less. However, good representation learning is crucial for the performance of many deep learning tasks, does it mean there would be inevitable performance v.s. forgetting tradeoff?

- The experiments seem to be of rather small scale: Although sufficient for supporting the theory, it would still be nice to see if the results can be transferred to real world applications.

**Questions:**

- What does the subscript k (y_k and x_k) mean in the equation at the bottom of page 3.

**Limitations:**

- It is unclear how the proposed interpretation can be generalized to more complicated networks or settings, e.g. deeper network or when the parameters move far away from the initialization.

---

> ### Author Rebuttal · Authors · 2024-08-07
>
> Thank you for your review and recognition of the novelty of our work.
>
> We would like to clarify some points raised under the weakness section before addressing specific questions. The first two bullet points raised the possibility that, although the infinite-width limit does not forget past tasks, this might not be overly surprising because networks in this limit do not learn much anyways. We partially agree, but in a more limited sense, as we describe below.
>
> Before responding, however, we want to emphasize that these are related to the general field-wide discussion about the utility of the NTK limit, rather than a comment on our specific contributions in results and insights. Even if this limit is not achieved, our results can provide useful insight. For example, without the proof that a neural network can be seen as an ensemble, one would not be able to describe the conditions under which the classifiers in this ensemble change. Even before the infinite-width limit, there are many strategies for ensuring this to be the case, such as moving in directions of low curvature or using smoother nonlinearities. These possibilities and many others are opened up by our perspective as normative goals for architecture and optimizer design, even if they are not perfectly achieved.
>
> >It is unclear to me, under the Neural Tangent Ensemble theory, how far can the model change from the initialization, if the model is not changing too much, then it is not surprising that the model is not forgetting, since it is not learning a lot information from the training data.
>
> We would like to slightly push back on the notion that in the NTK regime the network is not learning new information from the data. In the NTK and lazy training literature, it is true that the kernel is fixed in the infinite width limit. Yet there is still a significant learning problem, namely, to decide which kernel basis vectors are useful for a particular task. Similarly, in our ensemble interpretation, the learning problem is to decide which experts are useful. Quite a bit can be learned in this way if there are many experts. (Wider nets provide more functional diversity, mitigating the need to learn novel representations to perform well on a task.) To take a simple example, if the experts included all possible logistic regression classifiers, the ensemble learning problem is equivalent to Bayesian logistic regression. In general, the information learned from the data is the reduction in entropy over the ensemble, which may be significant. Our framework allows us to make this insight clear—there are weightings (i.e. small changes to initialization) that solve Task A well, and there are weightings that solve Task B well, but only the intersection of both weightings will solve both tasks well.
>
> >Similarly, the theory suggest that the model forgets less when getting wider, but the representation learning flexibility would also drop. So the model is learning less while forgetting less. However, good representation learning is crucial for the performance of many deep learning tasks, does it mean there would be inevitable performance v.s. forgetting tradeoff?
>
> We would like to slightly amend the tradeoff between continual learning and performance to a tradeoff between continual learning and *feature learning*. Yet, as we hold in the previous paragraph, this is not necessarily the same as performance. Our ability to learn without forgetting depends on the representational overlap between sequences of tasks. You are correct that when tasks have little representational overlap, we must trade-off between forgetting and performance. Yet, we stress that increasing the width/capacity of a network ameliorates this trade-off. Wider networks are likely to have more diverse experts, increasing the chance that the appropriate feature representations are available at initialization. More importantly, our theory provides a framework for understanding this trade-off, and our update rule is only one strategy for balancing it with the primary goal of mitigating forgetting. As we highlight in Section 5, the NTE framing suggests more sophisticated strategies for continual learning while limiting changes to the representations/experts. We believe that this work provides an understanding upon which the community can build such methods.
>
> >Q1: What does the subscript k (y_k and x_k) mean in the equation at the bottom of page 3.
>
> A1: The subscript k indexes over individual samples in the training dataset.

---

> > ### Comment · Reviewer_Gy8g · 2024-08-13
> >
> > I would like to thank the author for the detailed response! They resolved my concerns, I've increased my score.
> >
> > Overall I think this is a good paper that should be published, and I don't have too much concerns about the empirical evaluation's scale as I think the value of the paper is to provide a new framework for understanding continual learning.

---

### Author Rebuttal · Authors · 2024-08-07

We were happy to see that all 4 reviewers found our primary contribution – that linearized classifier networks are ensembles of experts (Theorem 1) – to be interesting, novel, well-supported, and relevant to problems of importance. Likewise, the reviewers appreciated the surprising nature and importance of our second theoretical result, which states that the posterior belief update rule for this ensemble is similar to SGD. The reviewers then noted several areas for improvement, and we believe the changes we have made in response have greatly strengthened the paper.

One common request was to clarify why an invariance to the ordering of data (thus solving continual learning) emerges when learning with an ensemble of fixed, independent experts. To address this, we have added a Lemma proving this fact. This can be found at the end of this general response.

Our empirical evaluations demonstrate that the ensemble provides useful insights that justify both known and novel observations about forgetting in NNs. However, all reviewers rightly pointed out that these evaluations were limited, leaving it unclear whether they would generalize to more complex networks and tasks. We have addressed this concern by adding additional continual learning tasks, datasets, and architectures. In particular, we evaluate a setup based on CIFAR-100 where each task is to predict a distinct subset of classes in the full dataset. We repeat our studies on modern convolutional networks such as ConvNeXt. Finally, we add additional experiments to address specific questions posed by reviewers, which we discuss in more detail in our individual responses.

Finally, we wish to further emphasize that our theorems provide a general result that will be useful for many subfields beyond continual learning. This work represents a new portal between two well-established domains. It enables a long history of theoretical results and strategies about mixtures of experts and ensembles to be applied to understand and improve neural networks. For example, this work provides a new path for uncertainty estimation, understanding bias/variance tradeoffs and generalization, sparsifying inference, and for architecture and initialization design—none of which we could fully investigate here.

Moreover, this result provides an intuitive starting point to reason about the transition from rich to lazy regimes. Rather than require the machinery of kernel regression, you can instead see neural networks instead as mixtures of experts. The rich regime corresponds to improving each expert, whereas the lazy regime corresponds to just weighing each expert but keeping them fixed. This understandable analogy can be taken in many directions, both for deep learning in theory and practice.

We believe these results are quite relevant for continual learning, which is why we framed this paper this way. Yet we are excited at the numerous open paths, both within continual learning and in other domains.

—---------------------------------------------------

Here we formalize the general statement that when a model class is a weighted ensemble of fixed, independent probabilistic classifiers, there is no catastrophic forgetting problem. This key fact motivates our assessment of under what conditions neural networks approach this setting.

When we update the weights of experts based on new data by Bayesian posterior updating, the end result is invariant to the ordering of data due to the multiplicative nature of probability updating. This is restated in the following Lemma, which we will include in the manuscript.

**Lemma: Invariance to data ordering in Bayesian Ensembles.**
Let $\mathcal{F} = {f_1, ..., f_N}$ be a set of fixed experts, $\mathcal{W}=w_1,...,w_N$ be their weights, and $\mathcal{D} = {D_1, ..., D_T}$ be a sequence of datasets from $T$ tasks. Let $w_i=p(f_i|\mathcal{D})$ be the posterior probability of expert $f_i$ given data $\mathcal{D}$.
Then, for any permutation $\pi$ of the indices {1, ..., T},
$p(f_i|\mathcal{D}) = p(f_i|D_1, ..., D_T) = p(f_i|D_{\pi(1)}, ..., D_{\pi(T)})$

*Proof*.
By Bayes' rule, $p(f_i|\mathcal{D}) \propto p(f_i) \prod_{t=1}^T p(D_t|f_i)$. The right-hand side is a product of terms, one for each dataset. Since multiplication is commutative, $\prod_{t=1}^T p(D_t|f_i) = \prod_{t=1}^T p(D_{\pi(t)}|f_i)$ for any permutation $\pi$. Therefore, $p(f_i|D_1, ..., D_T) = p(f_i|D_{\pi(1)}, ..., D_{\pi(T)})$.

---

### Decision · Program_Chairs · 2024-09-25

**Decision:**

Accept (spotlight)

**Comment:**

The paper has several interesting theoretical contributions, mainly new insights on interpreting a model as ensemble of classifiers and analyzing forgetting. The experimental design is interesting and further supports the theoretical findings. I recommend accept.